# One-Step Preparation of Nickel Nanoparticle-Based Magnetic Poly(Vinyl Alcohol) Gels

**Jun Li [1],\*, Kwang-Pill Lee [2,3] and Anantha Iyengar Gopalan [3]**

[1]   School of Pharmaceutical Sciences, Capital Medical University, Beijing 100069, China
[2]   Department of Chemistry Graduate School, Kyungpook National University, Daegu 41566, Korea; kplee@knu.ac.kr
[3]   Department of Chemistry, Kyungpook National University, Daegu 41566, Korea; agopalan@knu.ac.kr
\*   Correspondence: lijun88@ccmu.edu.cn

**Abstract:** Magnetic nanoparticles (MNPs) are of great interest due to their unique properties, especially in biomedical applications. MNPs can be incorporated into other matrixes to prepare new functional nanomaterials. In this work, we described a facile, one-step strategy for the synthesis of magnetic poly(vinyl alcohol) (mPVA) gels. In the synthesis, nickel nanoparticles and cross-linked mPVA gels were simultaneously formed. Ni nanoparticles (NPs) were also incorporated into a stimuli-responsive polymer to result in multiresponsive gels. The size of and distribution of the Ni particles within the mPVA gels were controlled by experimental conditions. The mPVA gels were characterized by field emission scanning electron microscope, X-ray diffraction, magnetic measurements, and thermogravimetric analysis. The new mPVA gels are expected to have applications in drug delivery and biotechnology.

**Keywords:** one-step preparation; nickel nanoparticles; magnetic poly(vinyl alcohol) gels

## 1. Introduction

In biological and pharmaceutical fields, polymers have been highly advanced by developing the synthesis methods, controlling the manufacturing steps, and designing the properties. Among such materials, hydrogels have been widely applied to meet versatile requirements [1]. Since PVA hydrogel was first formed by gamma rays in 1958, PVA hydrogels have been largely produced. Hydrogels were used in the biomedical and biotechnological fields in the 1990s. With high water content, elasticity, and biocompatibility, PVA hydrogels are widely used as biomaterials. However, there are limitations for PVA hydrogels: the characteristics of low permeability, stability, and fixation. In order to overcome the limitations, the properties of PVA hydrogels are enhanced by blending with other materials, such as other polymers, metals, and clays. Polyvinyl alcohols (PVA) are synthetic polymers widely used in industrial, medical, and food fields since the early 1930s [2]. PVA is relatively harmless when administered orally, because PVA does not accumulate in the body.

Magnetic metal nanoparticles have gained research interest in biomedicine for their chemical, electrical, and magnetic properties [3]. The crystalline structure, particle size, and magnetic properties of the magnetic metal nanoparticles can control and improve the polymers for applications in medical fields such as treatment of hyperthermia and drug delivery. The chemical composition and fabrication processes have great effects on the sizes and shapes of the particles. To achieve different properties, the synthesis routes are co-precipitation, sol-gel, ball-milling, gamma irradiation, laser irradiation, the photoinduced method, e1tc. The mediated polymers improve the quality of the particles for drug delivery. The idea of drug delivery was proposed by Paul Ehrlich [4]. To reduce the systemic distribution and the required drug dosage, magnetic carriers are used. In 1976, Zimmermann and

Pilwat proposed magnetic erythrocytes for the drug delivery. Since then, researchers began to use magnetic microparticles and nanoparticles (NPs) to target specific sites within the body. Some magnetic carriers are magnetic cores encapsulated in a biocompatible polymeric coating with drug loading capability. The magnetic cores are NPs of metal or metallic oxide moieties with sizes between 1 and 100 nm.

Magnetic gels aconsist of magnetic particles embedded within polymers [5]. The term is commonly applied to magnetic nanoparticles being immersed in a hydrogel. They are stimuli-responsive in an external magnetic field, but biological matter is tolerant to magnetic fields. They are promising candidates for controlled drug release due to the interplay between magnetic and elastic properties. The magnetogels combine the advantages of hydrogels and magnetic nanoparticles [6]. Hydrogels are similar to the cellular matrix with their high portion of water. Magnetic nanoparticles made of any transition metal (Fe, Ni, Co, Cr, or Mn) and its oxides allow for the control to a specific location under a magnetic field. The nanosystems improve target specificity, therapeutic effectiveness, magnetic resonance imageology, and hyperthermia for cancer therapy.

The magnetic drug carrier particles have been applied for over 40 years under an external magnetic field [7]. The magnetic nanoparticles can be removed to reduce risk of particle aggregation after therapy is completed. The nanoparticles can be synthesized in the presence of polyvinyl alcohol (PVA) or another substance to make them appropriate for in vivo applications. The polymers make the NPs stable [8,9]. One of the challenges is to improve the localization of cross-linker release to minimize in vivo toxicity. Controlling the density of cross-links in the gel matrix can tune the porous structure and the swelling properties of the hydrogels in the aqueous environment. Manufacturing processes still need to be developed for better-controlled hydrogels in terms of size, shape, distribution, and mechanics [10].

Ni-based nanomaterials can be used as magnetically-responsive therapeutic platforms for anticancer drugs [11]. Many different synthetic routes for magnetic nanoparticle synthesis have been reported [12] Some of them are one-step, while others are multi-step procedures. They all have advantages and disadvantages. But none of them provides a universal solution for all types of magnetic nanoparticles. Employing simpler and easier synthetic routes for the magnetic nanoparticles with desired characteristics remains highly challenging.

Doxorubicin (DOX) is a kind of cytotoxic anticancer drug. It is widely used in clinical therapy [13] To reduce the limitation of cardiotoxicity and improve the biocompatibility and efficiency, drug delivery systems are being developed [14]. The superparamagnetic nanodevice is designed to deliver DOX under a magnetic field, with good stability [15]. The main aim of this work is to design a new formation strategy of the magnetic polymer for the anti-cancer drug delivery system. An easy, one-step synthetic method for nickel nanoparticles based on poly(vinyl alcohol) gel is described. Nickel chloride was reduced by sodium borohydride in aqueous PVA solution. The Ni PVA gels were characterized by various techniques. The loading and release process was studied.

## 2. Experimental

### 2.1. Materials

PVA was purchased from Aldrich (Saint Louis, MO, USA) with a hydrolysis degree of 98%–99% and a molecular average weight of 85,000–124,000 g/mol. Nickel(II) chloride hexahydrated was from Junsei Chemical Co. (Tokyo, Japan). Sodium borohydride was from Kanto Chemical Co. (Tokyo, Japan). Doxorubicin hydrochloride (DOX) was obtained from Korea United Pharm. Inc. (Chungnam, Korea).

### 2.2. Synthesis of Ni-NP PVA Gel

Ni-NP PVA gel was prepared by reducing nickel chloride in PVA solution with sodium borohydride as the reducing agent. The aqueous solution of 10 wt.% PVA was first prepared by dissolving PVA in deionized water at 80 °C for 3 h. An aqueous solution of nickel chloride was then added to the PVA

solution in a beaker. $NaBH_4$ solution was added drop wise, using a separation funnel, to the nickel chloride solution, while the temperature was maintained between 80 °C with continuous stirring. The reaction mixture was allowed to stir for about 1 h at 80 °C, by which time black-colored solution was obtained, and at the same time a large amount of gas bubbles of $H_2$ were generated. The water in this solution was evaporated at 70 °C. After evaporation of most solvent, the hot residue was poured into a Petri dish. The black mixture was dried overnight at room temperature. After that, the residue was dried in an oven at 70 °C, and thus a gel was generated. The resulting gel was washed with distilled water, and further dried in an oven at 70 °C for 24 h. The resulting product was but to small pieces for various tests with scissors.

### 2.3. Characterization

The morphology and elemental composition of the structure, and microstructural characterization of the samples, were characterized using field-emission scanning electron microscopy (FE-SEM) (S-4200, Hitachi, Tokyo, Japan), operated at 3–10 kV electron potential difference and equipped with a semiconductor detector that allowed for the detection of energy dispersive X-rays (EDX). X-ray diffraction analysis (XRD) patterns on samples were measured on a model D8-Advanced AXS diffractometer (Bruker, Billerica, MA, USA) using Cu Kα radiation. Samples were supported on glass slides. Measurements were taken using a glancing angle of incidence detector at an angle of 2°, for 2θ values over 10°–80° in steps of 0.02°. Magnetic properties of the samples were measured by using a super-conducting quantum interface device (SQUID) magnetometer (MPMS-X 1, Quantum Design, San Diego, CA, USA) under an applied magnetic field, at room temperature. Thermogravimetric analysis (TGA) of the sample was performed on a TGA 7/DX Thermal Analyzer (Perkin-Elmer, Waltham, MA, USA) with a scan rate of 10 °C/min by pursuing $N_2$ gas as a carrier at a flow rate of 100 mL/min. The differential scanning calorimetric (DSC) experiment was carried out using a DSC2010 Differential Scanning Calorimeter (TA Instruments, New Castle, DE, USA) over a temperature range 20 to 1200 °C at a scan rate of 10 °C/min.

### 2.4. Swelling Studies

For the swelling kinetics' measurement, the gel (about 0.5 mm thick) was cut to 1 cm × 0.5 cm. The gel was then immersed in distilled water. The rate of gel expansion was determined by measuring the change in gel length at various time intervals. The dimensional change was estimated from the length ratio of the swollen hydrogel sample after 24 h (at the equilibrium) compared to its original length. The change in dimensions of the swollen samples ($L_s$) was calculated according to equation:

$$L_s(\%) = \frac{L_t - L_0}{L_0} \times 100\% \qquad (1)$$

where $L_0$ is the original gel length, and $L_t$ is the equilibrium gel length at a given time.

The water uptake of a gel sample was measured at room temperature. A swollen sample was removed from the solvent. After the water on the surface was absorbed gently with filter paper, the sample was immediately weighed. Then, the sample was returned to the medium. The mass swelling ratios ($M_s$) was calculated as follows:

$$M_s(\%) = \frac{m_t - m_0}{m_0} \times 100\% \qquad (2)$$

where $m_t$, $m_0$, and ($m_t - m_0$) are the masses of a swollen sample at a given time, dry gel, and absorbed water, respectively.

The equilibrium water content $W_e$ in swollen samples was calculated as:

$$W_e(\%) = \frac{m_\infty - m_0}{m_0} \times 100\% \qquad (3)$$

where $m_\infty$ is the weight of the swollen hydrogel at equilibrium [16].

### 2.5. Drug Loading and Release

The drug release kinetics is important for the application of the gel to drug delivery. Doxorubicin hydrochloride (DOX) was chosen as an anti-cancer model drug. The DOX loading was carried out by dispersing 10 mg of PVA gel in 3 mL DOX solution at a concentration of 0.02 mg/mL at room temperature in the dark. The mixture of PVA gel in DOX was shaken (80 rpm) at room temperature for 48 h to facilitate DOX uptake. The optical density of residual DOX in the supernatant was measured at 479 nm by UV-vis spectrophotometer (Cary 50, VARIAN, Palo Alto, CA, USA). The drug loading was determined as the difference between the initial DOX concentration and the DOX concentration in the supernatant. After the supernatant was removed, the release profile was obtained by reimmersing the gel loaded with DOX in 3 mL of water under gentle stirring. The concentration of DOX in the particle-free solution was determined at fixed time intervals by UV-vis spectrophotometry.

## 3. Results and Discussion

### 3.1. Reaction Mechanism of Ni-NP PVA Gel

Liquid phase reduction has some advantages over other synthetic methods for magnetic nanomaterials. Using some popular reducing agents, liquid phase reduction was applied to reduce magnetic metal ions to magnetic metal [12]. In this kind of reaction, hydrides are usually used for reducing agents. It is difficult to handle the sensitivity of hydrides to the mild environment. No special laboratory condition is required for these strong reactants, except for the moisture. Hydrides are also penetrative to some polymers, so the particles can still be reduced even with protective coatings.

NaBH$_4$ is a particularly powerful reducing agent in liquid phase reduction. It is soluble in both methanol and water. The mechanism of reduction for Ni using NaBH$_4$ is complicated. In the reaction, a black metal powder of Ni-NPs formed due to an instantaneous Ni$^{2+}$ → Ni co-reduction reaction with NaBH$_4$ [17]. The reduction of nickel ions using NaBH$_4$ followed the equation

$$NiCl_2 + 2NaBH_4 + 6H_2O \rightarrow Ni + 2B(OH)_3 + 2NaCl + 7H_2\uparrow$$

The byproducts of B(OH)$_3$ and NaCl remained dissolved in the aqueous solution and were removed after the washing process in fresh water.

Poly(vinyl alcohol) (PVA) is a good stabilizer for small metal particles in chemical synthesis to prevent the agglomeration and precipitation [18]. The embedding of the particles is also advantageous for the PVA casting. The Ni-NP PVA hydrogel was formed by the embedding of Ni-NPs into PVA aqueous solution. The Ni-NP remained in gel form within the matrix, as sketched in Scheme 1. Figure 1a,b shows photographs of the obtained PVA gel and Ni-NP PVA gel, respectively. Figure 1c,d shows photographs of the Ni-NPs and Ni-NP PVA gel synthesized by the same reduction reaction in the magnet field. When the Ni-NPs and the Ni-NP hydrogels were exposed to the field of a permanent NbFeB magnet, they were be attracted to the external magnet. This simple analysis demonstrated the magnetic properties of the polymers containing Ni-NPs.

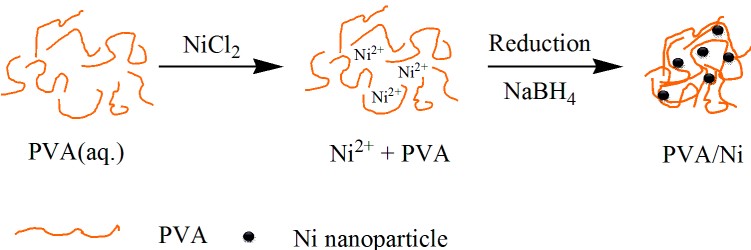

**Scheme 1.** One step preparation of PVA/Ni magnetic gel.

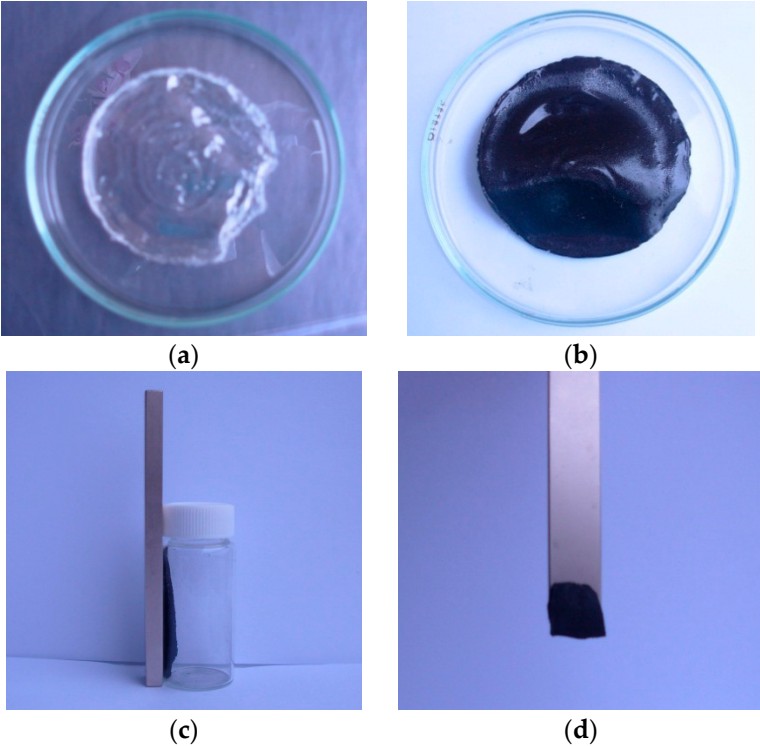

**Figure 1.** Photographs of (**a**) PVA gel; (**b**) PVA/Ni magnetic gel; (**c**) Ni-nanoparticle (NPs) in the magnet field; (**d**) PVA/Ni magnetic gel in the magnet field.

## 3.2. Structural and Morphological Properties

To explore the sample morphology, FE-SEM micrographs are presented. Figure 2a–f shows Ni-NPs, PVA gel, and PVA gel with magnetite particles. The surface of Ni-NP PVA gel is rough (Figure 2c) compared to the surface of PVA gel (Figure 2b). The neat PVA gel showed a very smooth surface. Figure 2c shows a micrograph of a Ni PVA gel with pores of about 50 nm. Ni-NPs' formation was evident from the presence of spherical particles over the PVA gel (Figure 2e) compared to the pure Ni-NPs formed by the same reduction reaction in aqueous solution (Figure 2a). From the image in Figure 2c, it can be seen that the magnetic particles are distributed in a compact PVA matrix. The EDX spectrum of PVA gel containing 2.3% Ni (Figure 2d) confirmed the presence of Ni-NPs. The EDX results revealed the percentage of Ni in PVA to be 10.97 wt.%. It is, thus, presumed that Ni-NPs have good dispersion in PVA gel. Compare the FE-SEM micrograph of PVA gel containing 2.3% Ni (Figure 2c) with 4.5% Ni (Figure 2e); the higher the amounts of magnetic nanoparticles, the rougher the surface of the Ni-NP PVA gel is. The EDX results (Figure 2f) revealed the percentage of Ni in PVA to be 41.15 wt.%.

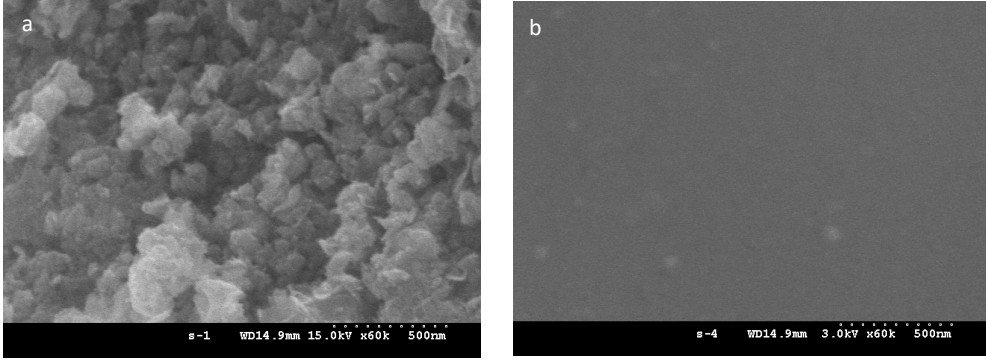

**Figure 2.** *Cont.*

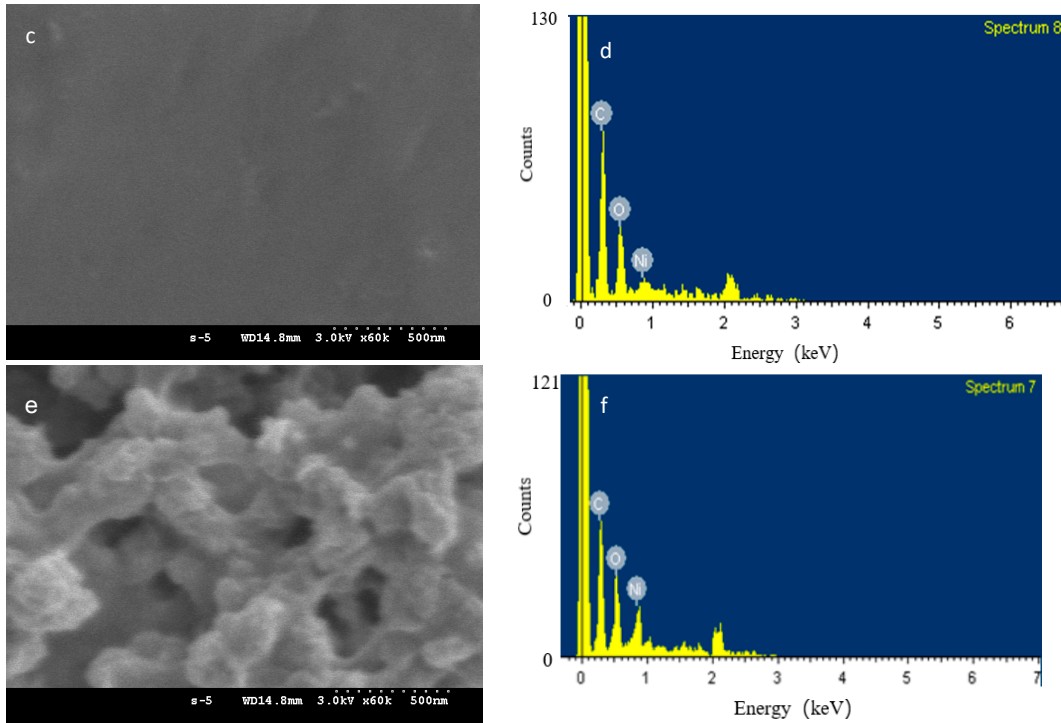

**Figure 2.** FESEM micrograph of (**a**) Ni-NPs; (**b**) PVA gel; (**c**) PVA gel containing 2.3% Ni; (**d**) EDX spectrum of PVA gel containing 2.3% Ni; (**e**) PVA gel containing 4.5% Ni; and (**f**) EDX spectrum of PVA gel containing 4.5% Ni.

Figure 3 shows the XRD spectra of PVA gel without or with Ni-NPs of different concentrations, and pure Ni-NPs obtained by the reduction reaction of nickel chloride with sodium borohydride. The XRD pattern of Ni-NPs (Figure 3d) shows that the products are metallic nickel. The average crystallite sizes were calculated from the peak broadening of XRD patterns by the Scherrer equation. The average size of the Ni-NPs was 7.0 nm. The X-ray diffractograms indicate a crystalline structure with some amorphosity of the disordered surface layers. The XRD spectra of Ni-NPs for 2θ values from 10° to 80° is similar to the literature, despite that an additional peak, the largest, at 2θ = 11.98° (intensity $I_p$ = 100 u), was not mentioned [19]. The XRD Patterns do not correspond to face centered cubic (fcc) nickel with space group Fm$\bar{3}$m, but are indexed for a tetragonal crystal structure with space group 14/mcm [17]. Those peaks at 2θ = 24.96° ($I_p$ = 7.0 u), 2θ = 34.06° ($I_p$ = 16.2 u), 2θ = 36.03° ($I_p$ = 3.7 u), 2θ = 45.04° ($I_p$ = 6.8 u), 2θ = 60.00° ($I_p$ = 8.7 u), and 2θ = 71.96° ($I_p$ = 2.2 u) are attributed to (110), (002), (200), (211), (310), and (321) facets of the tetragonal crystal structure of Ni, respectively. The lattice parameters are $a$ = 0.4887 nm and $c$ = 0.4516 nm for the nickel nanoparticle sample.

Figure 3a–c compares the X-ray diffractograms of PVA gel containing different amounts of Ni-NPs. The interplaner distance was calculated. XRD patterns of the pure PVA gel (Figure 3a) indicate the characteristic peak for poly(vinyl alcohol) at a 2θ value of 19.82 with the inter planner distance of 0.4476. The same phenomenon has been observed by Clémenson et al. [20]. PVA, known to be semicrystalline in nature, shows a single broad peak. The relatively sharp and broad peak centered on about 19 indicates that the semicrystalline nature of the polymer PVA contains crystalline and amorphous regions. It is clear from Figure 3b,c that the peaks representing the tetragonal crystal structure of Ni in XRD disappear, indicating the conversion of nickel nanoparticle to the amorphous Ni-NP PVA gel. For both PVA gel with 1.5 mol% Ni and 2.3 mol% Ni-NPs, a broad peak centered on a 2θ value of about 19 is observed. It suggests the characteristic peak of PVA gel, indicating that the Ni-NP PVA polymer intercalation occurs with the formation of Ni-NPs within the polymer matrix. The amorphous broad band decreased for PVA gel containing 2.3% Ni, indicating the metal interacts with the polymer chain [21]. From the Figure 3a–c, comparing the XRD spectra of the bulk Ni-NPs,

the smaller spectral intensity of the diffraction peak of PVA in the composite is due to the presence of a higher content of Ni-NPs in hydrogels.

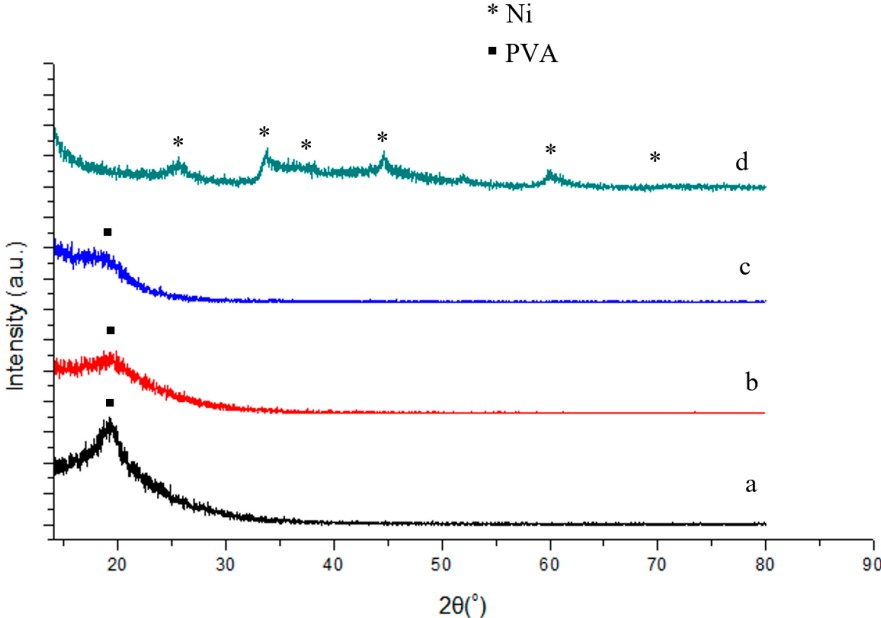

**Figure 3.** XRD patterns for PVA gel containing (**a**) 0% Ni; (**b**) 1.5% Ni; (**c**) 2.3% Ni; and (**d**) pure Ni nanoparticles.

### 3.3. Magnetic Properties

The magnetic properties of PVA gel with or without Ni-NPs of different concentration, and pure Ni-NPs were measured. Figure 4 shows the magnetization curves versus external magnetic field for neat PVA gel, or PVA gel with Ni-NPs of different concentration and pure Ni-NPs. Magnetization as a function of increasing external field up to 50 kOe at temperature of 5 K has been measured for selected samples. It can be found that the magnetization of pure Ni-NPs is obviously higher than other samples. The magnetic properties of Ni-NP PVA gel significantly changed with the composite formation. The magnetization of Ni-NP PVA gel is far weaker than that of the pure Ni-NPs. The greater the amount of Ni the Ni-NP PVA gel contains, the higher the value the magnetic moment shows at the same magnetic field. In contrast, the curve of the neat PVA gel is mostly flat. It is well known that the polymer is antimagnetic [22]. As shown in Figure 4, the curve of Ni-NPs shows that the magnetic moment does not reach full saturation even at 50 kOe. Among all these samples, the maximum magnetic moment value is 68.6 emu/g for pure Ni-NPs. The magnetization of the Ni-NPs and Ni-NP PVA gel samples increases quickly as the intensity of applied magnetic field increased below 10 kOe. The magnetization of the samples containing Ni increases slowly to reach saturation when the magnetic field above 10 kOe increases. The saturated magnetizations of the PVA gel containing 2.3 mol% Ni and PVA gel containing 4.5 mol% Ni at 50 kOe are 5.1 and 12.9 emu/g, respectively. Generally, the saturation of the magnetic moment increases with a higher amount of Ni-NPs. This is in agreement with the results of SEM and XRD.

In order to better understand the magnetic behavior, the samples were measured in the zero-field-cooled (ZFC) and field-cooled states (FC). The ZFC and FC measurements were performed by cooling the nanoparticles at zero field or in the presence of an external field. To obtain the ZFC measurement, the samples were cooled to 5 K in the absence of an external field. An external magnetic field of 1000 Oe was then applied to the samples. The magnetization was measured as the temperature was increased. The samples were then immediately cooled to 5 K in the presence of a magnetic field of 1000 Oe for the FC measurement. Figure 5a–c compares the magnetization versus temperature measured under ZFC and FC conditions for PVA gel without or with Ni-NPs of different concentration,

and pure Ni-NPs, respectively. The effect of cooling is clearly observed. ZFC and FC magnetization curves split below $T$ = 84, 54, and 35 K for PVA gels containing 2.3% Ni, 4.5% Ni, and the pure Ni gel, respectively. The difference in the magnetic moment signals is due to a two times larger amount of Ni in sample of PVA gel containing 4.5% Ni with respect to sample PVA gel containing 2.3% Ni, and the pure Ni without PVA. Samples of Ni-NP PVA gels and pure Ni nanoparticles present a significant magnetic irreversibility. The blocking temperature ($T_B$) of a magnetic nanoparticle can be measured where the FC and the ZFC curves diverge. The ZFC magnetization curve exhibits a peak around $T$ = 15 K. This temperature indicates a collective freezing of magnetic moments. The signal of a magnetic moment is relatively weaker when the amount of Ni in the sample is lower. On the other hand, no temperature dependence is observed for the PVA without Ni sample. It is confirmed that pure PVA is diamagnetic. The results establish the role of Ni loading within the PVA polymer for the magnetic moments. Figure 5a–c indicates that for all the samples containing Ni, the temperature dependence of the magnetic moment does not follow Curie's law at temperatures above 100 K. The difference between PVA gel and Ni-NP PVA gel suggests that magnetic response of the Ni particles manifests a certain degree of magnetic interaction [23].

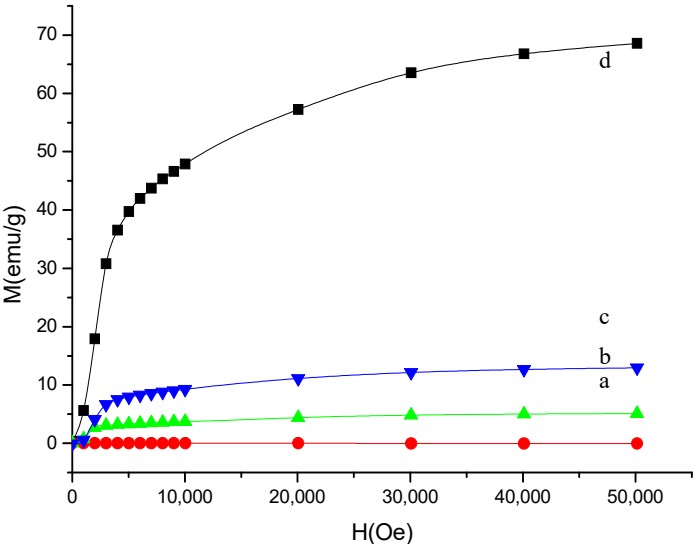

**Figure 4.** Magnetization curves of the PVA gel containing (**a**) 0% Ni; (**b**) 2.3% Ni; (**c**) 4.5% Ni; and (**d**) pure Ni nanoparticles.

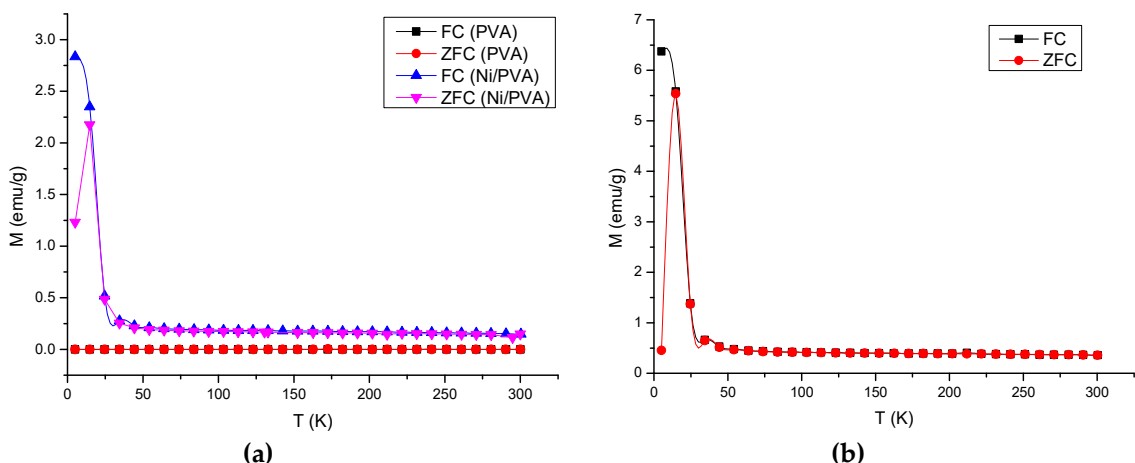

**(a)**　　　　　　　　　　　　　　　　　　　　　　**(b)**

**Figure 5.** *Cont.*

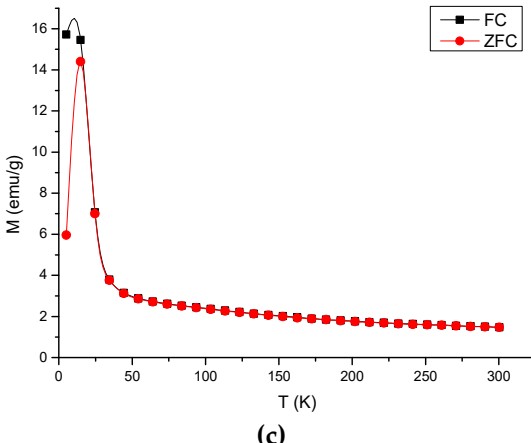

**(c)**

**Figure 5.** Field-cooling (FC) and zero-field-cooling (ZFC) magnetic moment curves measured for samples: (**a**) PVA gels containing 2.3% Ni; (**b**) 4.5% Ni; and (**c**) pure Ni nanoparticles. The lower curve of each sample represents the ZFC measurement and the upper one represents the FC measurement.

### 3.4. Thermogravimetric Analysis

To study the thermal changes in the PVA gels containing Ni, the samples were thermally characterized by TGA and DSC. A typical TGA curve of the PVA gel containing Ni is presented (Figure 6). The thermogram suggests that weight loss mainly occurs in three steps. First step corresponds to the loss of water molecules and slow removal of impurities from room temperature to 192 °C with a total weight loss of 14% in a PVA gel containing Ni. The second step corresponds to the removal of oligomers and the decomposition of PVA in the range of 266–294 °C with a weight loss of 29% [22]. The third step corresponds to the degradation of polymeric backbone in the range of 385–986 °C with a weight loss of 48% in the sample. At last, about 5% weight residue remained at a temperature above 986 °C, due to the Ni magnet. There is nearly no residue that remained above 986 °C in the PVA alone. This result is consistent with Khanna et al. [18]. Figure 6 shows the DSC thermogram of the PVA gel containing Ni [24].

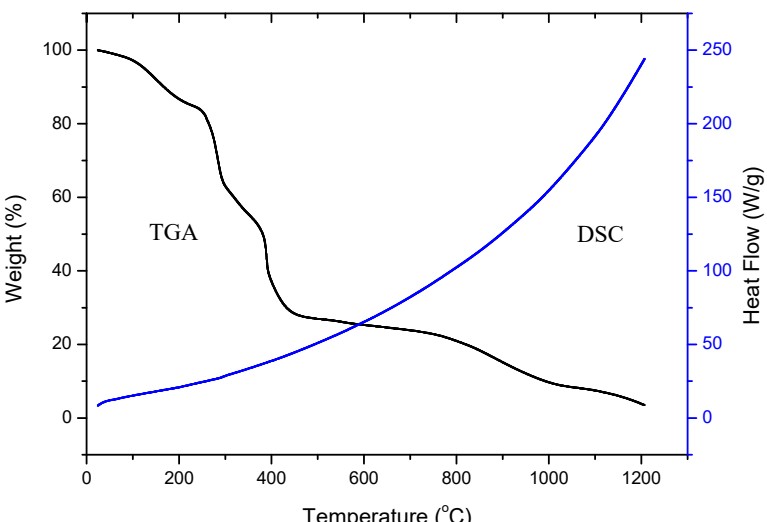

**Figure 6.** TGA curve and DSC plot of Ni-NP PVA gel.

### 3.5. Swelling Studies

Swelling experiments can yield important information concerning the stability of PVA gels in solution [25]. It can also be used to trigger drug release by controlling the hydrogel swelling properties. Water uptake was measured to determine the swelling abilities of the hydrogels. This is most commonly

used to evaluate the effects of different synthesis methods of PVA on water transport. The swelling kinetic curves of the hydrogels are shown in Figures 7 and 8. Figure 7 shows the dimension changes as a function of the exposure time for different PVA gels containing nickel in the amounts of 0 mol%, 1.5 mol%, 2.3 mol%, and 4.5 mol%. The mass change as a function of time from 5 min to 24 h for different hydrogels in water is shown in Figure 7. From Figures 7 and 8, the samples attained their maximum swelling ratio at about 16 h. The results of the dimension changes are in agreement with the results of the mass changes. The equilibrium water contents in swollen samples ($W_e$) were 364.60%, 312.36%, 230.16%, and 166.67% for neat PVA gel, and PVA gels with 1.5 mol%, 2.3 mol%, and 4.5 mol% Ni-NPs, respectively. From both Figures 7 and 8, it can be observed that the swelling ratio decreases as the amount of Ni-NPs in a PVA gel increases. The NPs have effect an on the swelling properties of the PVA gel by a simply physical method. The interaction of the PVA chains with Ni-NPs might induce the formation of low-mobility regions that can act as additional physical cross-linking points. The swelling ratio decreases with the presence of NPs, pointing to an increase of the physical crosslinking density as reported for the formation of PVA ferrogels through a freezing–thawing procedure [26].

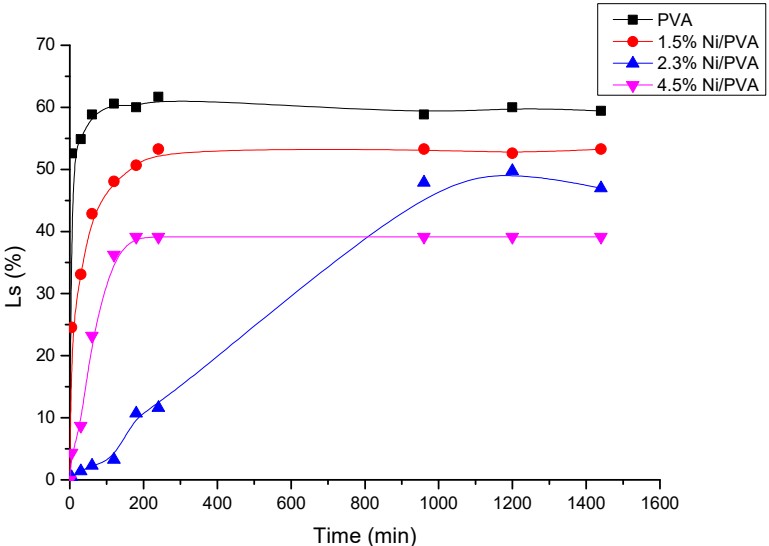

**Figure 7.** Dimension change ($L_s$) versus immersion time for PVA gel and PVA/Ni magnetic gel.

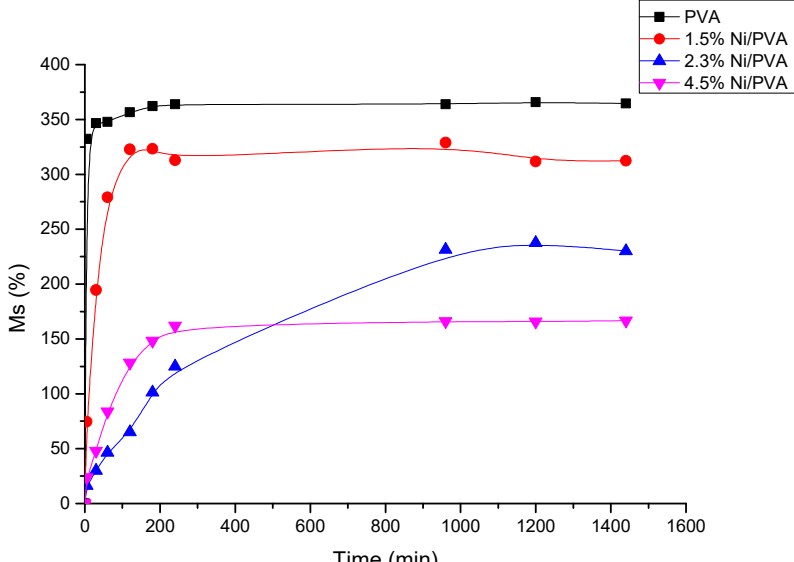

**Figure 8.** Water absorption ($M_s$) as a function of the exposure time for PVA/Ni magnetic gel.

### 3.6. Drug Loading and Release

To design the magnetic polymer for the anti-cancer drug delivery system, a well-known anti-cancer drug was needed. It was convenient to investigate the drug release properties in vitro. Furthermore, regarding swollen polymers—the drug is in an aqueous environment; its water solubility becomes an important consideration. As one of the most common anti-cancer drugs, DOX was chosen in this study to model the release action of anti-cancer drug from Ni-NP PVA gel. Most of the DOX was up-taken about in 24 h. A loading capacity of about 96% of the drug weight was determined. This result was used for the evaluation of the cumulative release of DOX. To investigate the drug delivery properties, Ni-NP PVA gels with a cargo amount of 5.8 mg/g were studied.

The release kinetics measured for Ni-NP PVA gel under two different conditions of 25 °C and 37 °C is shown in Figure 9. Both curves were measured by the UV-vis absorbance in water. From the figure, the curve with temperature of 25 °C shows a relatively low cumulative release of DOX. The corresponding cumulative release rapidly achieves the level of about 10% in 10 h. About 15% of the drug was released in 48 h. The release rate was much higher with the temperature of 37 °C. The corresponding cumulative release rapidly achieved the level of about 39% in 10 h. In total, 73% of drug was released in 48 h. This probably manifests through the fact that the Ni-NP PVA gel is made from simply physical cross-linking. This kind of gel is flexible when put in wet or very humid conditions under high temperatures. To study the release behaviors of the Ni-NP PVA gel, the cumulative release ratio ($M_t/M_\infty$) was calculated based on the classic Korsmeyer-Peppas equation [27]. $M_t/M_\infty$ is the fraction of drug released after time t relative to the amount of drug released at infinite time. There are two stages in the drug release curve. The initial diffusion of drugs mainly happened in the outermost layer of the Ni-NP PVA gel. At that stage, the drug released rapidly. At the second stage, the release was under diffusion's control and the drug released relatively slowly.

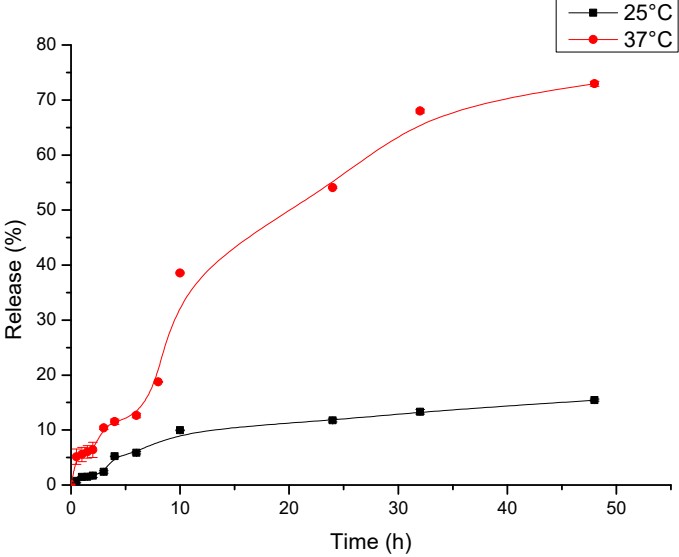

**Figure 9.** Drug release profiles of Ni-NP PVA gel in water.

## 4. Conclusions

The magnetic hydrogels have sparked particular interest in the anti-cancer drug delivery applications. PVA is a type of biocompatible material used for biomedicine. In this work, a new kind of nickel nanoparticle-based magnetic PVA gel was synthesized by a one-step procedure. The nickel nanoparticles and cross-linked mPVA gels were formed simultaneously. The structural and morphological properties of Ni-NP PVA gel were measured by FE-SEM and X-ray diffraction. The surface of Ni-NP PVA gel was rougher than that of the neat PVA gel. The amount of Ni-NPs detected was consistent with the theoretical value. The peak of PVA in the XRD spectra was weaker

than that of Ni-NP PVA gel. The magnetic and thermal properties were also measured. With a higher amount of Ni-NPs, Ni-NP PVA gel had higher magnetic moments. There are noteworthy features for the Ni-NP PVA gel in terms of ferromagnetism and thermally stability. All measurements confirmed the good formation of the Ni-NP PVA gel. The release of DOX showed diffusion control in vitro. With a higher temperature, the release rate of DOX was higher. The Ni-NP PVA gels are expected to be applied for controlled drug delivery. The novel, simple synthetic method can be used to form other magnetic gels for biotechnology.

**Author Contributions:** Conceptualization, K.-P.L.; methodology, A.I.G.; data curation, J.L.; writing—original draft preparation, J.L.; visualization, J.L.; supervision, K.-P.L.; project administration, K.-P.L. and A.I.G.; funding acquisition, K.-P.L.

**Funding:** This research received no external funding.

**Conflicts of Interest:** The authors declare no conflict of interest.

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
