# Peer review of "One-Step Preparation of Nickel Nanoparticle-Based Magnetic Poly(Vinyl Alcohol) Gels"

_coatings, doi:10.3390/coatings9110744_

Round 1

Reviewer 1 Report

In this paper a novel and easy one-step synthesis strategy for magnetic poly(vinyl alcohol) gels with nickel nanoparticles was developed. The gel was characterized by different techniques such as field-emission scanning electron microscopy, energy dispersive X-rays, thermogravimetric analysis and differential scanning calorimetry. Additionally, the swelling behavior of the gel under different nickel loading conditions was investigated. Finally, the applicability of the gel as a drug delivery system was analyzed trough observation of drug loading and release kinetics.

Some questions and remarks arose while reading the manuscript:

I strongly suggest that the manuscript is proofread, as there are many grammatical errors present in this version. I have added some examples (non-exhaustive) at the end of this review.

General remark: the introduction contains many repetitions. Some examples are: Page 2, lines 54- 58  is similar to the text on page 1, lines 35-36 and page 2, lines 69-70 is similar to the text on page 2, lines 51-52. What is also still missing is an overview of current formation strategies of PVA gels and how this synthetic method is different compared to state-of-the-art PVA gel synthesis methods.

Do the structural and morphological properties of the gel change in function of the nanoparticle amount? I can imagine that for higher amounts of magnetic nanoparticles, interactions occur and different links are built. This is sometimes observed in magnetic gels.

Page 2, line 78: Could the authors add a reference for the Doxorubicin drug?

Section 2.5: please introduce the DOX abbreviation

The ‘b’ is missing in Fig. 2. In Fig. 2b) the 500 nm inset is also missing.

In Fig. 5, I only see the results of the pure Ni nanoparticle case.

Page 7, lines 256-259: It was not clear to me why a deviation of Curie’s law would suggest MNP interactions (this might also be because of the missing curves in Fig.5)

Figs. 7 and 8: Do the authors have an idea why the sample with the 2.3% Ni, behaves so differently?

How do the drug release profiles adapt to other amounts of Ni in the PVA gel?

Figure 9: What is meant by ‘n = 3’?

What I feel is still missing in this paper is a comparison to other state-of-the-art magnetic gels, i.e. is stability similar, how do the drug kinetic profiles relate, etc.?

A list with some of the grammatical errors (not exhaustive) is presented below:

Page 1, line 27: Replace ‘are lack of selectivity’ with ‘lack selectivity’. The remainder of the sentence is not correct.

Page 1, line 39: This sentence is not correct.

Line 41: ‘which’ should be ‘with’, remove ‘are’.

Page 2, Line 52, ‘also can’ should be ‘can also’.

Page 8, line 274: This sentence is not correct.

Page 10, line 311: ‘in’ and ‘about’ should be switched.

Page 10, line 322: Positioning of ‘well’ is incorrect.

Reviewer 2 Report

Following issues must be addressed prior to publication:

General comments

the citation style throughout the manuscript does not match the guidelines the language, in particular the use of definite articles, throughout the manuscript should be improved

Introduction

citations should be added in the introduction, e.g. in the paragraph on page 2, lines 45-55

Results

how is the successful removal of by-products and residual educts of the Ni NP synthesis verified? Figure 2: reference in the text is mistaken with Figure 3 (page 5, line 172) mark (b) is missing in the Figure scale bars are poorly visible in all images and even overlapped in (a) overlapping of sub-figures must be corrected (d) is poorly edited what are the peaks at 2 and 0 (units are missing) in Figure (d)? Figure 3 + comment page 6, line 193: what is the strong peak then? why do amorphous Ni NPs form within the PVA matrix? This is an interesting observation and should be discussed in more detail! Figure 5 (a) and (b) are missing!! Figure 6: Tg at 276 °C is not convincingly visible (maybe due to scaling?) the discussion would significantly benefit form a comparison to pure PVA Figures 7 and 8 should be joint Why does the gel with 2.3% Ni behave completely different?

Reviewer 3 Report

The manuscript describes a one-step preparation of magnetic Ni nanoparticles dispersed in a PVA-based gel for biomedical applications (drug delivery). The nature and the size of the Ni NPs were characterized by SEM, X-ray, magnetic measurements and TGA-DSC. 

This is and overall interesting and multidisciplinary article, which offers some new perspectives for the preparation of magnetic gels. The manuscript could be suitable for publication in Coatings only after consideration of the points below:

1. The introduction needs partial re-writing. It can be confusing as it is very general, thus previous reports on similar systems should be reviewed in the introduction.

2. The authors only used one PVA polymer (Mw = 85-124 kDa). Did they try other molecular weights? What is the influence of the PVA molecular weight on the final properties of the gels?

3. TEM studies would be very useful (and more accurate than SEM) to determine the size of the Ni NPs. 

4. According to XRD patterns, Ni peaks were not detected. Did the authors try to prepare gels with higher concentrations of Ni NPs?

5. The TGA curve showed several weight losses and the authors explain why their materials lost weight at such temperatures. How did they assigned them? Did they use a TGA-IR or TGA-MS? In addtion, the authors reported a Tg of 276ºC by DSC, however it looks like that the materials are decomposed at such temperature. Why?

Round 2

Reviewer 3 Report

In the revised version, the authors have addressed most of the issues that I raised, and the present manuscript does have a significant improvement over the first submission. Therefore, I now suggest acceptance.